# CA1 and CA3 differentially support spontaneous retrieval of episodic contexts within human hippocampal subfields

Halle R. Dimsdale-Zucker [1,2], Maureen Ritchey[3], Arne D. Ekstrom[1,2], Andrew P. Yonelinas[1,2,4] & Charan Ranganath[1,2]

The hippocampus plays a critical role in spatial and episodic memory. Mechanistic models predict that hippocampal subfields have computational specializations that differentially support memory. However, there is little empirical evidence suggesting differences between the subfields, particularly in humans. To clarify how hippocampal subfields support human spatial and episodic memory, we developed a virtual reality paradigm where participants passively navigated through houses (spatial contexts) across a series of videos (episodic contexts). We then used multivariate analyses of high-resolution fMRI data to identify neural representations of contextual information during recollection. Multi-voxel pattern similarity analyses revealed that CA1 represented objects that shared an episodic context as more similar than those from different episodic contexts. CA23DG showed the opposite pattern, differentiating between objects encountered in the same episodic context. The complementary characteristics of these subfields explain how we can parse our experiences into cohesive episodes while retaining the specific details that support vivid recollection.

[1] Center for Neuroscience, University of California, Davis, CA 95618, USA. [2] Department of Psychology, University of California, Davis, CA 95618, USA. [3] Department of Psychology, Boston College, Chestnut Hill, MA 02467, USA. [4] Center for Mind and Brain University of California, Davis, CA 95618, USA. Correspondence and requests for materials should be addressed to H.R.D.-Z. (email: hrzucker@ucdavis.edu)

Considerable evidence suggests that the hippocampus is essential for episodic memory and that it plays a particular role in binding information about items and the context in which they were encountered[1,2]. Most mechanistic models suggest that the hippocampal subfields play complementary roles in spatial and/or episodic memory[3–7]. Although these models generally predict large differences between neural coding in CA1 and CA3, between-subfield differences at the level of single units in rodents and in overall activity in human fMRI studies have been relatively modest. Indeed, both CA1 and CA3 have been implicated in representations of temporal[8,9] and spatial[10,11] contextual information. One suggestion[12] is that the different computations supported by CA3 and CA1 should be most apparent when one analyzes the population-level activity patterns elicited by different contexts—whereas CA3 should differentiate between specific experiences in the same context and CA1 should globally differentiate between different contexts.

One way to do this is with pattern similarity (PS) analyses. We can think of PS analyses as capturing the activity across voxels that act as spatiotemporal filters sampling from blood vessels, which, in turn, respond to fluctuations in activity from large populations of neurons[13]. Voxel PS analysis is analogous to population vector analysis approaches used in single unit recordings (e.g., ref. [14]). If we assume that activity in each voxel reflects the outcome of vascular sampling of neural responses from large, distributed populations, then the voxel pattern is a filtered, macro-scale analog of the neural population vector. Just as different population vectors reflect changes in the relative firing rates across different neurons across two conditions, different voxel patterns may reflect changes in the underlying population-level response across two conditions.

Here, we used high-resolution fMRI and multivariate analysis methods to test how different hippocampal subfields contribute to representations of spatial and episodic context. We designed a virtual reality environment consisting of two houses (spatial contexts; Fig. 1). After becoming familiarized with the spatial layouts of each house, participants viewed a series of 20 videos (episodic contexts) depicting first-person navigation through each house while they encountered a series of objects. Each object was studied only once and was uniquely placed in a single house within a single video. Following this study phase, we scanned participants while they performed an item recognition test that required them to differentiate between studied and novel objects. Although the items were displayed without any contextual information, based on cognitive models of recognition memory and models of human hippocampal function, we predicted that recollection-based item recognition should trigger reactivation of information about the context in which that item was encountered[1,2]. Accordingly, we tested whether multi-voxel patterns elicited during item recognition carried information about the associated spatial (house) or episodic (same house and video) context.

## Results

**Behavioral results**. Behavioral data are presented in Supplementary Table 1. Recognition memory performance was indexed by evaluating responses to new and old items in the object recognition test completed during MRI scanning. Accuracy was high for both hits ("remember" hit rate = 0.68 [SD = 0.17]; "familiar" hit rate = 0.25 [SD = 0.15]) and correct rejections (correct rejection rate = 0.90 [SD = 0.04]). False alarms were not present for the majority of participants ($N = 17$ with no "remember" false alarms; $N = 3$ with no "familiar" false alarms), therefore we did not compute discriminability indices. Remember rates varied by spatial context, $t(22) = 3.33$, $p = 0.003$, $d = 0.21$, such that recognition was slightly lower for items that had been studied in the gray house.

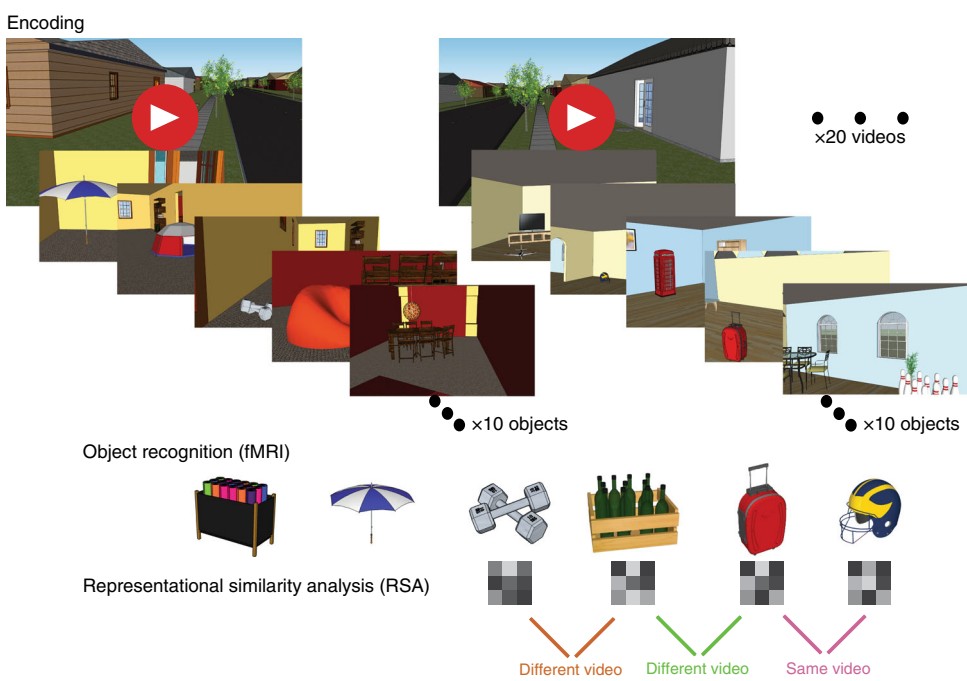

**Fig. 1** Experimental approach. Participants encoded objects uniquely located within one of two spatial locations (spatial contexts) across a series of 20 videos (episodic contexts). Next, they were scanned while performing an object recognition test which required differentiating old and new objects presented without any contextual information. We used representational similarity analyses (RSA) to examine the similarity of voxel patterns elicited by each recollected object relative to other recollected objects that were studied in the same (or different) spatial and episodic context. This figure is not included under the Creative Commons licence for the article; all rights reserved

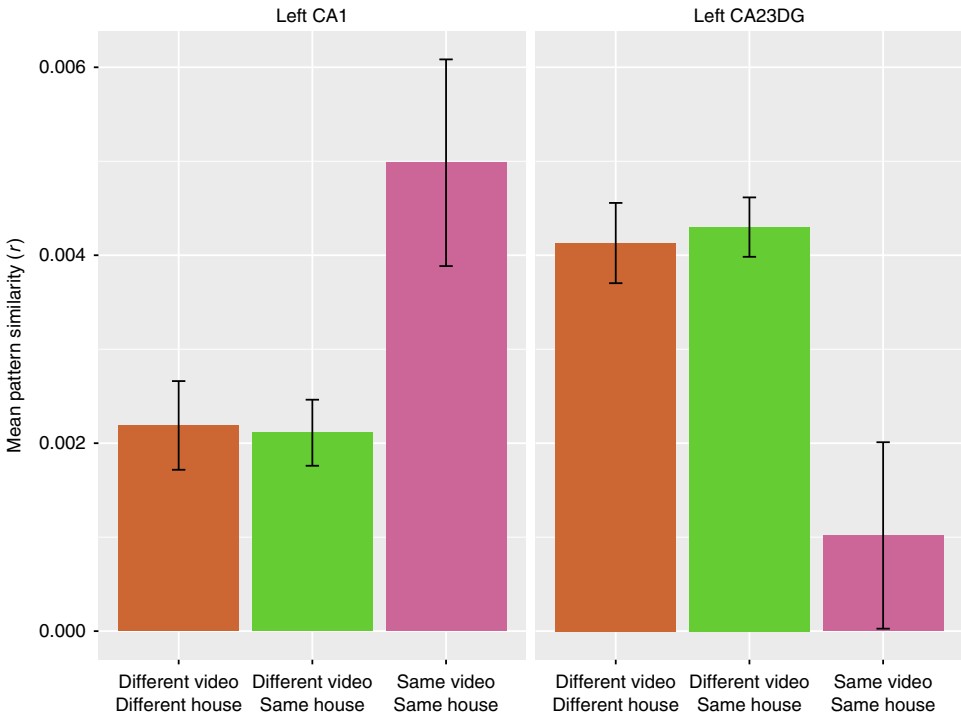

**Fig. 2** Pattern similarity in CA1 and CA23DG is sensitive to episodic context. Pattern similarity was higher in left CA1 for items studied in the same video (Same Video Same House) than for items in different videos (Different Video Same House). Left CA23DG showed a reversal of this pattern such that pattern similarity was higher for items studied between videos vs. within the same video. Neither CA1 nor CA23DG patterns were sensitive to spatial context similarity alone

Because participants' responses in the scanner were based on subjective measures of recollection, we directly assessed memory for the spatial context associated with each item in a test administered immediately after MRI scanning. Overall, spatial context memory was high (mean spatial context memory hit rate = 0.71 [SD = 0.11]; Supplementary Table 1), and did not reliably vary between the two spatial contexts, $t(22) = 1.06$, $p = 0.30$, $d = 0.19$. Consistent with the idea that item recollection involves activation of associated information, memory for spatial context was higher for items endorsed as remembered than for those endorsed as familiar, $F(1, 22) = 74.72$, $p < 0.001$, $\eta_p^2 = 0.51$.

**fMRI results**. Many standard-resolution and high-resolution fMRI studies have reported enhanced hippocampal activity during successful recollection and/or retrieval of spatial source information (for a review, see ref. [15]). Consistent with these prior studies, whole-brain voxel-wise comparisons revealed increased activity in the left hippocampus during recollection hit trials as compared to activity during familiarity hit trials and misses (FWE corrected $p < 0.05$; Supplementary Fig. 2a, Supplementary Table 2). Estimated hemodynamic response functions for each subfield for this contrast as well as univariate estimates of subfield activity for each spatial context can be found in Supplementary Figs. 3 and 4, respectively. Outside of the hippocampus, activation in medial prefrontal cortex and other regions in the "core recollection network"[16] was increased during recollection trials as compared to familiar hits and misses (Supplementary Fig. 2B). These results suggest a general role for the hippocampus in item recollection in addition to the differential roles the subfields play to support memory for contextual information.

Having established hippocampal recruitment for item recollection, we proceeded to investigate whether hippocampal activity

patterns carried information related to spontaneous retrieval of spatial and episodic contexts for these recollected trials.

As shown in Fig. 1, we estimated single-trial multi-voxel patterns within regions of interest (ROIs) corresponding to CA1 and a combined CA2/CA3/dentate gyrus (CA23DG) subregion within the body of hippocampus. Distributions of single-trial activity estimates are included in Supplementary Figure 6 and representative PS values are shown in Supplementary Figure 7. Specifically, we computed voxel PS between trial pairs for recollected items that shared the same episodic context (i.e., same video/same house), shared the same spatial context but with different episodic contexts (different video/same house), or were associated with different episodic and spatial contexts (different video/different house). To maximize the likelihood of identifying trials that were associated with successful context retrieval, we restricted analyses to trials that were associated with correct recollection-based item recognition and correct identification of spatial context (house) in the post-scan context memory test.

To test whether regions carried information about an item's encoding context (spatial/house, episodic/house and video), we fitted a mixed model with a random effect of subject[17] testing for effects of ROI (CA1, CA23DG), context similarity (same episodic, same spatial, different context), and hemisphere (left, right), as well as their interactions on PS values. There was a significant ROI × context similarity × hemisphere interaction ($\chi^2(2) = 13.30$, $p = 0.001$). Follow up analyses revealed that this was driven by a reliable interaction between ROI and context similarity in left ($\chi^2(2) = 15.64$, $p < .001$) but not right ($\chi^2(2) = 1.65$, $p = 0.44$; see Supplementary Figure 5). To further break down this interaction, we conducted separate analyses restricted to left hemisphere in our ROIs to assess representation of context similarity.

To investigate whether regions carried information about an item's spatial encoding context (house), we compared PS values for items that had been studied in the same house or different

house. In the same house condition, we eliminated trial pairs that had been studied within the same video to ensure that any observed effects could uniquely be attributed to spatial context similarity and not episodic context similarity. Based on traditional models[3,18], we expected to see greater PS for same-house as compared to different-house pairs in CA23DG but not in CA1. As can be seen in Fig. 2, neither CA23DG ($\chi^2(1) = 0.08$, $p = 0.78$) nor CA1 ($\chi^2(1) = 0.03$, $p = 0.86$) systematically differed in their representation based on an item's spatial context. These results indicate that neither CA1 nor CA23DG were differentially sensitive to spatial context similarity alone.

To investigate whether activity patterns in hippocampal subfields carried information about episodic context, we compared PS values between pairs of trials that were studied in the same video (which necessarily meant that that the items had also been studied in the same spatial context) against PS values for pairs of trials that were studied in different videos that depicted the same spatial context. Some models[3,12] predict that CA1 should treat items from the same episodic context as more similar to one another than CA23DG. Indeed, in CA1, activity patterns were more similar across pairs of items from the same episodic context than across pairs from different videos ($\chi^2(1) = 6.50$, $p = 0.01$). Intriguingly, CA23DG showed the reverse pattern; that is, PS was significantly lower for items in the same video than for items in different videos ($\chi^2(1) = 10$, $p = 0.002$). We observed a significant interaction, ($\chi^2(1) = 15.45$, $p < 0.001$; Fig. 2), indicating that the PS profiles of CA1 and CA23DG were qualitatively different for episodic context similarity.

A control analysis was performed to ensure that our results could not merely be explained by the number of trials in each condition since the same video/same house condition necessarily had the smallest number of trial pairs contributing to the PS analysis. To do this, we took the condition with the fewest number of trial pairs on a subject-by-subject basis and randomly selected a matching number of trial pairs from all other conditions. For example, if a participant had 152 trial pairs in the same-video/ same-house condition, we randomly selected 152 trial pairs for all other conditions by generating a random sequence of numbers of length 152 and then using this random sequence to index trial pairs for selection in the mixed modeling of PS. We repeated this simulation 1000 times and looked to see whether our observed $\chi^2$ values for the critical interaction of ROI × context similarity × hemisphere interaction were significant at a threshold of $p < 0.05$ (Supplementary Figure 9). Across nearly all of these random samples of trials, the effect was statistically significant.

We next investigated whether the effects of episodic context on PS were driven by just a few influential voxels. To better characterize the nature of the observed patterns, we repeated our analyses dropping five influential voxels in three different ways. First, we identified voxels that consistently had the largest absolute magnitude of response across all trials. These voxels could drive the PS result by increasing the magnitude of the observed correlations. Second, we identified voxels that had the largest standard deviation in their response across all trials. These voxels could contribute to PS differences due to their variability, thereby driving the observed between-condition differences. Third, we identified voxels that had the largest mean squared differences between trial pairs in the same video/same house and different video/same house conditions. We defined voxel variability relative to these two conditions since we observed a significant difference in the ROIs only in the episodic context condition. These voxels could lead to inflated PS and the observed between-condition differences due to systematic variability between the contexts. Across all three approaches, dropping influential voxels did not change the observed pattern of results.

An additional control analysis was performed to ensure that differences in PS on the basis of context similarity did not simply reflect differences in reaction times between conditions[19]. As suggested by Todd et al.[19], we re-modeled our data including a random effect of the reaction time difference in trial pairs that went into the PS analyses. Results remained unchanged after removing the effect of reaction time differences for all comparisons of interest (Supplementary Figure 10).

To show the specificity of these effects to CA1 and CA23DG, we repeated our PS analyses using entorhinal cortex and subiculum instead (Supplementary Figure 8). These regions are in the hippocampal formation, but not considered to be subfields of the hippocampus proper. For episodic context similarity, neither entorhinal cortex (left: ($\chi^2(1) = 0.09$, $p = 0.77$; right: ($\chi^2(1) = 0.51$, $p = 0.47$) nor subiculum (left: ($\chi^2(1) = 0.15$, $p = 0.70$; right: ($\chi^2(1) = 3.75$, $p = 0.053$) differed in their representation based on an object's episodic context. For spatial context similarity, neither entorhinal cortex (left: ($\chi^2(1) = 0.64$, $p = 0.42$; right: ($\chi^2(1) = 1.79$, $p = 0.18$) nor subiculum (left: ($\chi^2(1) = 0.15$, $p = 0.70$; right: ($\chi^2(1) = 0.08$, $p = 0.77$) were differentially sensitive to spatial context alone.

## Discussion

Our results reveal striking differences in retrieval of contextual information across the hippocampal subfields and provide a rare statistical dissociation between CA1 and CA23DG. In CA1, activity patterns were more similar across trials that involved recollection of the same episodic context (same video) than across trials that involved retrieval of different episodic contexts (different videos); in CA23DG, PS was lower between trials that were associated with the same episodic context than between trials that were associated with different episodic contexts. These results are consistent with the idea that CA1 represents global contextual regularities across items ("pattern completion"), whereas CA23DG exaggerates differences between items that have competing associations within the same episodic context ("pattern separation"). Together, CA1 and CA23DG can play complementary roles in supporting episodic memory by allowing one to remember specific items, as well as their relationships, within a shared context.

Differences between the subfields are a prominent component of models of subfield function[3,12], which, in turn, are based on anatomical differences in the inputs, connections, and firing properties of the subfields[20]. Studies in rodents have reported differences between spatial coding in CA1 and CA3 following changes in the environment[21,22], when learning spatial tasks such as the watermaze[23], in distinguishing non-spatial sequences[24], and in contextual fear conditioning paradigms[25]. However, we are not aware of any prior findings in humans that have reported dissociations between the roles of CA1 and CA23DG during spontaneous retrieval of contextual information.

One previous human fMRI study reported a statistical dissociation between CA1 and CA23DG as participants monitored the spatial layouts of cities that varied in similarity[26]. Stokes et al. found that CA23DG represented cities with the same layout as more similar than cities with different layouts whereas CA1 was not sensitive to the change in layouts[26]. One critical difference between their study and ours is that Stokes et al.[26] directly assessed memory for spatial layouts, whereas we assessed incidental retrieval of contextual information during item recognition. Our results challenge traditional notions that CA3 is particularly sensitive to changes in spatial contexts[12,26] in that PS values were not sensitive to spatial context information alone. The findings indicate that, although space may be important in defining an episodic context, spatial context alone does not

account for differences between representations in CA1 and CA23DG during memory retrieval.

The present results converge with other findings[10] showing that, when spatial information alone is insufficient to resolve context, spatial information does not drive representations within CA3. Our findings also accord with Leutgeb and Leutgeb's[12] proposal that CA3 can differentiate between different sensory cues that are encountered in the same place.

Our findings dovetail with extant evidence that CA1 plays a critical role in representing time[8,27–29], in representing sequences of ordered information[24,30–36], and in using this temporal information to define episodes[37–39] under demands that mimic a realistic, real-world episodic context. Additionally, our findings support the idea that CA1 is critical for distinguishing similar episodic contexts[3], given that we saw greater PS in CA1 for items within the same video as compared with those that occurred between videos.

The fact that CA23DG showed lower neural similarity for items encountered during the same video, as compared to items encountered during different videos, seems at odds with theories proposing a critical role for CA23DG in episodic memory retrieval[3]. Examined more closely, however, the results align with recent findings indicating that the hippocampus differentiates between related information in an episode[40–43]. For instance, building on the theory of Marr[18], it has been argued that dentate gyrus and CA3 work together to distinguish related information (i.e., "pattern separation"; for a review see ref. [44]). Our finding that CA23DG is more likely to individuate objects within a video—resulting in lower neural similarity—is consistent with the idea that CA3 generally pushes apart representations of similar items (see also refs. [45–47] for related findings).

Another possibility is that the coding scheme of CA23DG may be flexible based on the elements that have priority in the task[4]. In some tasks, such as those involving spatial navigation, distinguishing between competing spatial representations may be essential to correct navigation and subsequent memory performance[46], and, thus, CA23DG may be pushed toward orthogonalization. Successful performance on our task requires one to differentiate between representations of items encountered in the same video that share a spatial environment. However, if we could construct a task in which there were lower demands to orthogonalize item-specific features, we might expect CA23DG to show a representational scheme more consistent with pattern completion. That is, we would expect CA23DG to show increased similarity for items in the same context relative to items encountered in different contexts.

In one recent demonstration of the flexibility of coding schemes in CA23DG, Aly et al.[48] asked participants to deploy attention either to objects within a context (pieces of art) or to the context itself (the layout of the room). Critically, stimuli were identical between the conditions but participants' attentional state varied based on the task demands. They found that CA23DG, but not CA1, showed a greater match in its profile of activity to the task-relevant attention state. This task-driven modulation of activity was related to increased subsequent memory for incidentally encoded information[48]. In a related study, they replicated the finding that attention state-specific patterns of activity in CA23DG were uniquely related to subsequent memory for the studied items[49]. Such findings indicate that the coding scheme in CA23DG can change according to task demands, and this can have consequences for later memory performance.

Both temporal and spatial details are key defining features of episodic memories[50], but, until now, it was not clear how these spatiotemporal contexts are represented by the hippocampus during retrieval. CA1 and CA23DG exhibited activity patterns that were sensitive to retrieved episodic context information in

the absence of memory differences for the objects themselves. The complementary characteristics of these subfields explain how we can parse our experiences into cohesive episodes while retaining the specific details that support vivid recollection[7,51–53].

## Methods

**Participants**. Twenty-eight participants took part in the study. Of these, one failed to complete their MRI session due to technical malfunctions with the scanner. An additional four subjects were excluded due to not having at least two usable runs of fMRI data (either due to excessive motion, $N = 2$, exiting the MRI scanner between runs, $N = 1$, or low behavioral performance resulting in correctly recollected trials only in a single run, $N = 1$). Analyses presented are from the remaining twenty-three participants ($N_{female} = 11$, mean age = 19.5 years). The study was approved by the Institutional Review Board of the University of California, Davis, and all participants provided written informed consent at the time of the study.

**Stimuli and materials**. Study materials included two virtual homes created in Google SketchUp (https://www.sketchup.com/, version 15.3.329). Homes were matched for total virtual square area and subdivided into two rooms. Each house contained ten pieces of landmark furniture that shared semantic labels (e.g., "couch") but differed in appearance (e.g., angular gray couch vs. plush green couch) between the two homes. Exterior color, wall color, room orientation, and decoration style also differed between houses (Supplementary Fig. 1).

Three-hundred neutral objects (e.g., football helmet, suitcase, teddy bear, vase, and phone booth) were selected from the Google SketchUp image library. Two-hundred and forty of these objects were randomly selected and positioned within the homes in sets of ten objects. Object assignment to homes was random and was house- and video-unique so that 12 lists of objects were assigned to each home. To determine object placement, rooms were divided into eighths, all possible combinations of five positions were generated, and objects were randomly assigned to one of these combinations. Thus, object configurations within each room (and thus within each house) were video-unique.

Videos depicting trajectories through the houses were generated using Google SketchUp's animation feature. Videos were exported for each house both with landmark furniture only and for each of the 12 videos with objects within each home. Trajectories did not change between videos within a home. Each video was ~1 min 40 s in duration.

**Procedure**. After providing informed consent, participants practiced the four phases of the task using an example video recorded in the experimenter's (HDZ's) house. Phase one consisted of context familiarization, phase two object encoding, phase three object recognition (scanned), and phase four location recognition.

In the context familiarization phase, participants passively viewed videos of the two homes with landmark furniture but devoid of any other objects. Androgynous names ("Alex", "Jamie") were randomly assigned to each house and a label with the owner's name appeared above the video for all presentations (e.g., "Jamie's House"). After watching the video tours of both Alex and Jamie's houses twice, participants were presented with a blank map that included rectangles where the houses were located and the street between the houses but no other contextual details. Maps were generated for the purposes of strengthening participants' representations of the houses (spatial contexts). Participants were given up to 10 min to draw maps of each house including the location of doors, walls, furniture, and the owner's name associated with each house to ensure thorough knowledge of the spatial layouts. Before progressing to the object encoding phase, the experimenter reviewed the accuracy of their maps using a transparency overlay and corrected any mistakes.

Following context familiarization, participants progressed to the object encoding phase. During this phase, participants viewed a series of 20 videos. Each video depicted passive navigation through one of the two homes, with ten different objects placed along the trajectory. At the end of each video, a still frame of each of the ten objects in situ in the house appeared one at a time in random order for 4 s while the participant judged whether the object was worth more than $50 (yes/no). Responses were recorded via keypress but were not analyzed. After making this value judgment, the still frame of the object was replaced by a birds-eye-view perspective map of the house where each room was divided into quadrants with numeric labels. Participants indicated the quadrant number where the object had been located during the video via keypress (1–8). Again, responses were recorded but not analyzed as the purpose of these encoding judgments was to solidify participants' memory both for object identities and their locations. In total, participants saw ten videos in each house presented such that the order of the houses alternated across even and odd videos (e.g., house1/video1, house2/video2, house1/video3, house2/video4, etc.). The ten videos were randomly selected from the pool of 12 videos in each house and the order of video presentation was uniquely randomized for each participant. This phase took roughly 1 h to complete.

The object recognition test phase took place across four runs in the MRI scanner following acquisition of structural scans. Sixty-three still images were presented in each run for 3 s with a jittered inter-trial interval ranging from 2 to 8 s. All 200 studied objects were presented with an additional 52 new, unstudied objects. New objects were randomly selected from a pool of objects that were not presented in either house. While the image was on the screen, participants made

recognition judgments ("remember", "feels familiar", "new") via an MRI-compatible button box. Participants were instructed to make a "remember" response when they could bring to mind a specific detail from when they had studied the object, "feels familiar" if they thought they had studied the object but were unable to retrieve a specific detail, and "new" for objects that they did not think they had studied during object encoding. Critically, no house or video information was re-presented to participants in these object images. Following four runs of the object recognition task, participants additionally completed a spatial localizer task in which they saw short videos (15 s) and still frames from both houses with landmark furniture but no objects. The results of this scan will not be discussed further here.

During the final phase of the experiment, participants were brought back to the lab to complete a spatial context memory test. In this phase, participants were re-presented with the 200 studied objects, and they were asked to recall where (house and room) each object had been studied (Alex room 1, Alex room 2, Jamie room 1, Jamie room 2; names were presented in the same orientation as the houses). Images remained on the screen for 3 s while participants made their response. Data from this phase were used to ensure that item memory did not differ as a function of spatial context.

**fMRI acquisition and pre-processing**. Scans were acquired on a Siemens Skyra 3T scanner with a 32 channel head coil. Two sets of structural images were acquired to enable subfield segmentation: A T1-weighted magnetization prepared rapid acquisition gradient echo (MP-RAGE) pulse sequence image (1 mm$^3$ isotropic voxels) and a high-resolution T2-weighted image (TR = 4200 ms; TE = 93 ms; field of view = 200 mm$^2$; flip angle = 139°; bandwidth = 199 Hz/pixel; voxel size = 0.4 × 0.4 × 1.9 mm; 58 coronal slices acquired perpendicular to the long-axis of the hippocampus). High-resolution functional (T2*) images were acquired using a multiband gradient echo planar (EPI) imaging sequence (TR = 2010 ms; TE = 25 ms; field of view = 216 mm; image matrix = 144 × 152; flip angle = 79°; bandwidth = 1240 Hx/pixel; partial phase Fourier = 6/8; parallel imaging = GRAPPA acceleration factor 2 with 36 reference lines; multiband factor = 2; 52 oblique axial slices acquired parallel to the long-axis of the hippocampus slices; voxel size = 1.5 mm$^3$ isotropic).

SPM8 (http://www.fil.ion.ucl.ac.uk/spm/) was used for image pre-processing. Functional EPI images were realigned to the first image and resliced. No slice timing correction was performed due to the acquisition of multiple simultaneous slices with the multiband sequence. Co-registration between the native-space ROIs defined in T2 space and the functional images was done with SPM's Coregister: Estimate and Reslice procedure. This procedure uses the normalized mutual information cost function between a reference (mean functional) image and source (T2) image to compute and apply a voxel-by-voxel affine transformation matrix. This transformation matrix was then applied to the subfield ROIs that had been defined in T2 space (see 'ROI segmentation') to bring them into register with the functional images (Supplementary Figure 11). The T1 image was co-registered to the mean EPI. Then, nonlinear spatial normalization parameters were derived by segmenting the coregistered T1 image. For the voxel-wise univariate analyses, these normalization parameters were applied to the contrast images to bring all images into MNI space before group analysis. Contrast images were additionally smoothed with a 3 mm FWHM Gaussian kernel. Quality assurance (QA) included identifying suspect timepoints via custom code (https://github.com/memobc/memolab-fmri-qa) defined as time-points in excess of 0.5 mm frame displacement (based on[54] or 1.5% global mean signal change (based on ARTRepair recommendations[55]). Runs were excluded if the frame displacement exceeded the voxel size. Two participants were excluded for motion in excess of these thresholds and one run was excluded from another participant based on these criteria.

**ROI segmentation**. Hippocampal subfield ROIs were segmented using the ASHS toolbox[56] with the UPenn atlas. Segmented ROIs were co-registered to the mean functional image and split into masks for each ROI of interest. We took a conservative segmentation approach in which we combined subfields DG, CA3, and CA2 into a combined region based on prior high-resolution fMRI work[26,57,58] despite ASHS segmentations for each subfield. Head, body, and tail were manually defined but subfield comparisons were limited to the body where boundaries between CA1 and CA23DG can be most clearly and reliably delineated[59]. Disappearance of the gyrus intralimbicus was used as the head/body boundary[70] and presence of the wing of the ambient cistern demarcated body/tail[61].

**Data analysis**. ROI summary statistics, including PS analyses, were computed using custom code implemented in MATLAB r2014b (www.mathworks.com) and R version 3.3.2 (http://www.R-project.org). Statistical comparisons were conducted in R including linear mixed models implemented with lme4[62]. Our mixed models included fixed effects of condition, ROI, and hemisphere as well as a random subject intercept. P-values were obtained by comparing a full model with the effect of interest against a reduced model without this effect and quantified with Chi-squared likelihood ratio tests[17,60,63–65]. Cohen's $d$ effect sizes for within-subject paired samples $t$-tests were computed correcting for correlation between means as suggested by Dunlap et al.[66].

**Univariate fMRI analyses**. Six conditions of interest (remember and familiar hits split by house, correct rejections, false alarms) were modeled with event-related stick function regressors. Six motion parameters and additional spike regressors as identified by our QA scripts were also included in the model. Trials of no interest (e.g., where no response was made) were included as a nuisance regressor. "Recollection" trials were defined as trials on which a subject correctly endorsed remembering an item that had been presented during encoding. "Familiarity" trials were defined as trials on which a subject correctly endorsed having seen an item but did not endorse being able to remember any specific details about its initial presentation[67]. Because there were not enough familiarity trials to analyze this condition separately, comparisons of interest combined familiar hits with miss trials.

Contrast maps for recollection-related activity were computed by comparing the activation difference for recollection hit trials as compared to the combined activity for familiarity hit trials and misses. T-maps were cluster corrected at FWE < 0.05 using a combined voxel-wise threshold ($p < 0.001$) and cluster extent ($k = 88$).

**PS analyses**. PS analyses[13] were conducted on beta maps generated from unsmoothed data in native subject space. Following the procedure described by Mumford[68,69], single trial models were generated to estimate the unique beta map for every trial in a run ($N = 63$). Within each single trial model, the first regressor modeled the trial of interest with a stick function, the second regressor modeled all other trials in that run, six regressors were used to capture motion, and any additional spike regressors as identified by our QA scripts were used to capture additional residual variance. Following beta estimation, outlier values were identified by setting a $z$-scored beta threshold of 0.7–0.85 based on visual inspection of the distribution of $z$-scored beta values for all subjects. This resulted in an average of 9.87% (mean = 6.22 trials, SD = 7.10 trials) excluded beta values per run for each participant.

Voxel-wise patterns of hemodynamic activity were separately extracted for each ROI from the single trial beta images. Within each ROI, correlations (Pearson's $r$) were computed between these trial-wise betas to yield a trial-by-trial correlation matrix that related each voxel's signal on a trial to all other trials across all runs. We restricted comparisons to those trials for which participants made a correct "remember" response (during MRI scanning) and correctly identified an item's spatial (house) context (post-MRI spatial source task). Statistical analyses tested for differences in correlations between trial pairs on the basis of encoding context (same vs. different house, same vs. different video within a house). Only between-run correlations were used to maximize the number of possible trial pairs without mixing within- and between-run correlations. Trial pairs of interest were extracted from these trial-by-trial correlation matrices. For all conditions, we restricted our trial pairs to those where participants made both a correct remember response as well as a correct house source memory judgment.

**Data availability**. Minimally processed data needed to regenerate analyses and figures can be found online (https://osf.io/5th8r/) as well as relevant analysis code (https://github.com/hallez/abcdcon_pub). All other data that support the findings of this study are available from the corresponding author upon reasonable request.

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

## Acknowledgements
We are grateful to the members of the UC Davis Memory Group for their valuable feedback and support throughout the design, analysis, and writing phases. This work was supported by NSF-GFRP 1650042 (H.R.D.-Z.).

## Author contributions
H.R.D.-Z., M.R., A.D.E., A.P.Y., and C.R. designed the study and wrote the paper. H.R.D.-Z. performed the experiment and analyzed the data.

## Additional information

**Competing interests:** The authors declare no competing financial interests.

