## [Peer Review File · Nature Communications]

Reviewers' comments:

Reviewer #1 (Remarks to the Author):

This is a novel and interesting report that dissociates hippocampal areas CA1 and CA3 (actually CA2/3/DG) in operations that can be considered pattern completion and pattern separation, respectively. The study was largely well designed (but see below), the analyses proper, and the findings convincing. The results extend our understanding of different hippocampal subdivisions by revealing differences in actual memory task, unlike previous work by Stark and others whose findings are based on passive viewing of same vs different images unrelated to memory performance. I do, however, have some concerns that need to be addressed.

1. I am confused about what happens in each video. A key sentence in the methods seems to suggest that subjects view "20 unique episodes with house order alternating" (p. 14) – do the houses alternate within each episode or does each episode involve just the same house the episodes alternate in the house used? The main text (bottom p.7) seems to indicate that each video (= episode?) involves only the same house. It would have been useful to show different houses in the same video/episode in order to complement all the other video and house combinations and examine episode similarity alone with house/space randomized. In particular, this would have been useful to dissect out the role of house/space in contributing to episodic context (see #4 below).
2. I am confused about the use of the term "encoding context" and "context" on p. 6. Do they mean episodic context or spatial context, or what? More generally, it is quite confusing that they mix references to "video" and "episode", as well as between "episodic context" and "spatial context" and the various video/house combinations. They should make all of this very clear at the outset and use consistent terms thereafter.
3. While the dissociation in Fig 2 is very clear, I note that the level of similarity for spatial coding alone (different video/same house) is low in CA1 and higher in CA23DG. Is this a real difference indicating a distinction in spatial processing across areas?
4. Also in Fig 2, doesn't the observation that space alone (different video/same house) does not differ from unrelated (different video/different house) suggest this test did not detect any spatial coding by either hippocampal area? In other words, space contributes at most only as a part of episode coding (same video/same house), right (referred to obliquely in next to last paragraph on p. 9)?
5. A general issue with dissociations of CA1 and CA2/3/DG is that the output of the latter circuitry is mainly to the former (CA1) and not to cortex, and otherwise CA3 only projects to subcortical areas. So the main output of the hippocampus for memory processing in the cortex is only the pattern completion version of hippocampal analysis. In light of this, what is the role of CA23DG pattern separation?

Reviewer #2 (Remarks to the Author):

Dimsdale-Zucker and colleagues describe a very interesting experiment to test whether CA1 versus CA23DG hippocampal subfields are differentially involved in recollection of spatial versus episodic context information. They presented subjects with videos of objects embedded within two different spatial contexts (houses) and later tested memory for the objects individually. This was an advantageous design feature because any fMRI differences between conditions of interest could therefore be attributed to the content of recollected contextual information rather than to any stimulus-specific effects. fMRI pattern similarity (PS) analysis performed within each subfield showed that CA1 PS was maximal among items within the same episodic context (video) whereas CA23DG PS was maximal among items with different episodic and spatial contexts. This does not seem to fit with the initial hypothesis of different spatial versus episodic contextual memory functions of these subregions, but does provide evidence consistent with the idea of distinct pattern completion versus separation functions of CA1 versus CA23DG. Although the exact nature of the connections between this finding and the existing literature on CA1 versus CA23DG distinctions could be greatly improved, this finding is potentially very important because it demonstrates the first functional dissociation between CA1 and CA23DG in humans (previous papers have shown differential sensitivity, but not full statistical dissociation).

The major weakness of this paper is the superficial presentation of the data/results. The full fMRI analysis of the primary conditions (not counting the recollection/familiarity analysis that is peripheral) is presented in only one summary bar graph of PS values (Figure 2). So much is missing that it makes it difficult to evaluate the weight of the evidence in this figure. Below I'll summarize a few of the missing details, but in general this paper would be strengthened by including a thorough presentation of the findings, including potential new control analyses, so that the reader can fully evaluate whether the CA1/CA23DG statistical distinction is convincing. A secondary weakness is that ties with the previous literature on hippocampal subfield specialization is also somewhat limited, and should be expanded and clarified in order to better contextualize the novelty of this finding. It is of course the case that space is limited, but including additional details even in supplementary materials would be sufficient.

Specific suggestions:

PS values should be shown for the right hippocampal subfields as well. It is misleading to run an analysis that includes hemisphere as a factor, but then only show the data for the hemisphere that happened to show a statistical effect. It makes me wonder what is going on with the other hemisphere (very noisy, same pattern but less reliable, etc.), and some answers to that question could indicate problems with the data.

A related issue is that a control region would have been a nice addition (a non-hippocampal location) to show specificity of these effects to hippocampal subfields.

Page 8 says "Control analyses that matched the number of trials across conditions revealed consistent results". This is an alarming statement because it both raises the possibility that

the main findings could be due entirely to different trial counts across conditions and gives a vague answer to that possibility. Absolutely no details are given about the nature of these control analyses or what they showed. How is the reader to evaluate this? What was done and what counts as “consistent results”?

The main analysis of subfield fMRI PS really needs to be complemented by a corresponding univariate analysis in order to allow evaluation of the data quality and therefore the quality of the PS findings. What is going on under the hood? They should show estimated impulse response functions for the hippocampal subfield ROIs. The PS was run on estimated betas, which could vary based on how well the canonical HRF matches the subfield response, which could bias the PS analysis for subfields with relatively canonical versus relatively non-canonical responses. Indeed, at least one of the references supporting the idea of subfield specialization indicates different CA3/CA1 timecourses of activity (Lee, Rao and Knierim 2004). Perhaps different activity timecourses are responsible for the reported analyses? At very least, the beta estimates of the subfields that were analyzed should be given so the reader knows at least how well the model fit the data for these regions.

Ideally, univariate contrasts would be performed among the same conditions that were submitted to PS analysis. If there are univariate activity differences among the same conditions that show PS distinctions, then this would raise the possibility that PS effects are simply reflections of different univariate activities, which changes the interpretation. Mean univariate activity could be controlled in the PS analysis if needed.

The accuracy and response time data should be given for the conditions of interest for the PS analysis, not just nonspecifically for correct/incorrect trials and response type (currently Supplementary Table 1). It is already suggested that accuracy varied among conditions of interest for the PS analysis, and presumably response times could have varied as well. Given existing data that factors like response time variation can yield spurious classification results (e.g, Todd et al. 2013 Neuroimage), including full control PS analysis that explicitly address accuracy and response-time variation among condition should be performed.

The methods for segmenting subfields and co-registering fMRI data to these subfields leaves much to faith. The methods section should include relevant data/images to show accurate segmentation and alignment of functional data.

What is the nature of the “patterns” captured by the PS analysis? Overall, mean PS is strikingly low ($\sim r=0.003$ on average from Figure 2). What does this mean? Are the differences in PS among conditions due to just one or two voxels being more alike/different in one condition versus the other? Given the geometry of the hippocampal subfield masks, what do different patterns reflect? There is not much space in the coronal plane, so do different patterns indicate differences in activity along the long axis? Some unpacking of the nature of PS values, including potential visualization of similar/different patterns of representative PS levels, would be very useful in understanding the nature of the reported effects.

The rationale for the current univariate analysis is unclear. Why run a univariate analysis of

recollection-related activity when the main PS comparisons involve differences by context condition? The point seems to be to show that the task “engages” the hippocampus, but why showing this for memory conditions unrelated to the conditions of interest is important is unclear (one could presumably look at things like all stimuli versus baseline to show “engagement”). If the authors keep the current univariate analysis, activity should be summarized in table form to allow thorough evaluation and to facilitate meta-analyses.

One would have guessed that right hippocampus would be important for this task, but all findings were specific to left. Univariate responses for recollection also seem to be on left (Figure 1, although details on orientation of display seem missing so hard to tell if it is left or right). Was the laterality unexpected? Should additional guards against multiple comparisons be used given that left hippocampus does not seem to be an a priori hypothesis.

The rationale for not performing slice-timing correction given MB acquisition is unclear. It is possible.

A rationale for why mixed modeling was used rather than traditional repeated-measures ANOVA should be included. It seems like a simple 2x3x2 design, so the rationale for modeling is not clear.

Response to reviewers:

Reviewers' comments:

Reviewer #1 (Remarks to the Author):

This is a novel and interesting report that dissociates hippocampal areas CA1 and CA3 (actually CA2/3/DG) in operations that can be considered pattern completion and pattern separation, respectively. The study was largely well designed (but see below), the analyses proper, and the findings convincing. The results extend our understanding of different hippocampal subdivisions by revealing differences in actual memory task, unlike previous work by Stark and others whose findings are based on passive viewing of same vs different images unrelated to memory performance. I do, however, have some concerns that need to be addressed.

1. I am confused about what happens in each video. A key sentence in the methods seems to suggest that subjects view “20 unique episodes with house order alternating” (p. 14) – do the houses alternate within each episode or does each episode involve just the same house the episodes alternate in the house used? The main text (bottom p.7) seems to indicate that each video (= episode?) involves only the same house. It would have been useful to show different houses in the same video/episode in order to complement all the other video and house combinations and examine episode similarity alone with house/space randomized. In particular, this would have been useful to dissect out the role of house/space in contributing to episodic context (see #4 below).

We thank the reviewer for raising this need for clarification in the description of what participants encoded. Each video (“episode”) involved just one house and the episodes alternated in the house used. We have now changed the text on page 17 to the following such that it is now consistent with the description of the task in the main text on page 7:

[Again, responses were recorded but not analyzed as the purpose of these encoding judgments was to solidify participants’ memory both for object identities and their locations.] In total, participants saw 10 videos in each house presented such that the order of the houses alternated across even and odd videos (e.g., house1/video1, house2/video2, house1/video3, house2/video4, etc.). The 10 videos were randomly selected from the pool of 12 videos in each house and the order of video presentation was uniquely randomized for each participant. This phase took roughly one hour to complete.

The reviewer also highlights that it would have been useful to use different houses within the same episode to allow us to disentangle episodic (video) similarity from spatial (house) similarity. This is an excellent idea, but it is not clear whether that design would fully disentangle episodic and spatial similarity, because participants might segment different sections of the same video into different events (for a review, see Radvansky and Zacks, 2011). Crossing spatial boundaries (e.g., the street between the houses in our virtual neighborhood) is known to reliably trigger event segmentation, such that people differentiate between memories for experiences from the two spatial contexts (e.g., Radvansky and Copeland, 2006). Therefore, had we used multiple houses in the same episode, this could have introduced variability in how participants defined episodes.

Understanding how the neural representation for items within the same spatial location or different spatial locations that are presented within the same video remains an interesting question that is open for future work. Our prediction would be that if participants represent the different locations as within the same episode (i.e., house + car + garage = getting ready episode) then we would again see greater similarity for within-episode as compared to across-episode items within CA1 whereas CA3 would pick up on the item-unique features. However, if participants were to think of these separate locations as distinct episodes then we would only see greater similarity in CA1 for items in the same spatial location (e.g., house/house items > house/car items or house/garage items).

2. I am confused about the use of the term “encoding context” and “context” on p. 6. Do they mean episodic context or spatial context, or what? More generally, it is quite confusing that they mix references to “video” and “episode”, as well as between “episodic context” and “spatial context” and the various video/house combinations. They should make all of this very clear at the outset and use consistent terms thereafter.

On page 6, this sentence has been clarified as follows (added text underlined, now page 7):

To test whether regions carried information about an item’s encoding context (spatial/house, episodic/house and video), we fitted a mixed model with a random effect of subject testing for effects of ROI (CA1, CA23DG), Context Similarity (same episodic, same spatial, different context), and Hemisphere (left, right), as well as their interactions on PS values.

We now use the terms “spatial context” and “episodic context” throughout the remainder of the paper for consistency. Additionally, we now use the term “video” rather than “episode” to further eliminate confusion. These changes are highlighted in the main text of the revision.

3. While the dissociation in Fig 2 is very clear, I note that the level of similarity for spatial coding alone (different video/same house) is low in CA1 and higher in CA23DG. Is this a real difference indicating a distinction in spatial processing across areas?

The reviewer raises an important theoretical argument about whether there is an overall difference in spatial coding across the subfields. As with other fMRI approaches, raw pattern similarity values for a single condition cannot be interpreted directly—to identify a distinction in spatial processing, it is necessary to compare pattern similarity differences between pairs of items that share a spatial context (same house) versus those that do not (different house). Neither subfield showed a significant difference between these conditions, and a Spatial Context (Same House vs Different House) x Subfield (CA1 vs CA23DG) analysis revealed no significant main effects or interactions. Thus, the results do not indicate a difference in spatial context representation across the areas during memory retrieval. However, it is worth noting that Stokes et al. (2015) reported differences in spatial processing across the subfields during virtual navigation, and we point readers to this finding on p. 12.

4. Also in Fig 2, doesn't the observation that space alone (different video/same house) does not differ from unrelated (different video/different house) suggest this test did not detect any spatial coding by either hippocampal area? In other words, space contributes at most only as a part of episode coding (same video/same house), right (referred to obliquely in next to last paragraph on p. 9)?

We agree with the reviewer that our observation that spatial context alone (different video/same house vs. different video/different house) didn't lead to systematic differences in PS in either subfield does seem to suggest that space contributes at most only as a part of the episodic context. We have strengthened our description of this on page 12 to the following:

Our results challenge traditional notions that CA3 is particularly sensitive to changes in spatial contexts (Leutgeb & Leutgeb, 2007; Stokes et al., 2015) in that pattern similarity values were not sensitive to spatial context information alone. The findings indicate that, although space may be important in defining an episodic context, spatial context alone does not account for differences between representations in CA1 and CA23DG during memory retrieval.

5. A general issue with dissociations of CA1 and CA2/3/DG is that the output of the latter circuitry is mainly to the former (CA1) and not to cortex, and otherwise CA3 only projects to subcortical areas. So the main output of the hippocampus for memory processing in the cortex is only the pattern completion version of hippocampal analysis. In light of this, what is the role of CA23DG pattern separation?

The reviewer brings up an interesting and important theoretical question – if CA23DG is feeding pattern separated information into CA1 but CA1 is feeding a pattern *completed* representation out to cortex, then what is the influence of CA23DG on memory? Although our data cannot directly speak to this, we can consider some possible roles for pattern separation in CA23DG.

First, it is possible that CA23DG influences the neocortex via polysynaptic connections. For instance, we know that CA3—and CA1—projects to the lateral septal nucleus which then projects out to supramammillary nucleus which projects to neocortex (Risold and Swanson, 1996). This is just one example of other potential routes for CA23DG to share its pattern separated output more widely than just to CA1.

Second, it is possible is that, even though population-level activity patterns in CA1 differ from CA23DG, these responses might still depend on pattern separation in CA23DG. CA1 receives input from ERC layer 3 as well as from CA3. CA3 and DG, in turn, receive input from ERC layer 2 (Aggleton and Christiansen, 2015; Witter et al., 2000). Thus, CA1 can be thought of as integrating two somewhat different sources of information. Consequently, it is possible that CA1 *requires* CA23DG's pattern separated interpretation of its unique information to facilitate recovery of contextual information that is associated with a specific item.

Third, it is possible that the differences that we observed between CA1 and CA23DG during retrieval do not necessarily reflect what occurred during encoding. For instance, it is possible that CA1 and CA23DG exhibited patterns that diverged only late in learning (i.e., as participants

encoded more items in each context) or during the post-learning period. Computational models also support the idea that CA1 and CA3 learn differently (Schapiro, Turk-Browne, Botvinick, and Norman, 2016) and have different roles across encoding and retrieval (Ketz, Morkonda, and O'Reilly, 2013).

We did not present these ideas in the paper, because we felt that they were too speculative, but we can do so if the reviewers feel that it would be helpful.

Reviewer #2 (Remarks to the Author):

Dimsdale-Zucker and colleagues describe a very interesting experiment to test whether CA1 versus CA23DG hippocampal subfields are differentially involved in recollection of spatial versus episodic context information. They presented subjects with videos of objects embedded within two different spatial contexts (houses) and later tested memory for the objects individually. This was an advantageous design feature because any fMRI differences between conditions of interest could therefore be attributed to the content of recollected contextual information rather than to any stimulus-specific effects. fMRI pattern similarity (PS) analysis performed within each subfield showed that CA1 PS was maximal among items within the same episodic context (video) whereas CA23DG PS was maximal among items with different episodic and spatial contexts. This does not seem to fit with the initial hypothesis of different spatial versus episodic contextual memory functions of these subregions, but does provide evidence consistent with the idea of distinct pattern completion versus separation functions of CA1 versus CA23DG. Although the exact nature of the connections between this finding and the existing literature on CA1 versus CA23DG distinctions could be greatly improved, this finding is potentially very important because it demonstrates the first functional dissociation between CA1 and CA23DG in humans (previous papers have shown differential sensitivity, but not full statistical dissociation).

The major weakness of this paper is the superficial presentation of the data/results. The full fMRI analysis of the primary conditions (not counting the recollection/familiarity analysis that is peripheral) is presented in only one summary bar graph of PS values (Figure 2). So much is missing that it makes it difficult to evaluate the weight of the evidence in this figure. Below I'll summarize a few of the missing details, but in general this paper would be strengthened by including a thorough presentation of the findings, including potential new control analyses, so that the reader can fully evaluate whether the CA1/CA23DG statistical distinction is convincing. A secondary weakness is that ties with the previous literature on hippocampal subfield specialization is also somewhat limited, and should be expanded and clarified in order to better contextualize the novelty of this finding. It is of course the case that space is limited, but including additional details even in supplementary materials would be sufficient.

We thank the reviewer for raising these concerns about the brevity of the presentation of the results. As we mentioned in the cover letter, this manuscript was initially transferred from Nature Neuroscience as a Brief Communication. We conducted many analyses that were omitted from the manuscript due to the strict word and reference limit for the Brief Communication format.

Our revised manuscript takes advantage of the additional space in the Nature Communications format to include several analyses that address Reviewer 2's concerns:

1. Estimated FIR for each subfield, Supplemental Figure 3; response to comment #4
2. Mean univariate activity for each subfield broken down by spatial context, Supplemental Figure 4; response to comment #4
3. PS in left and right hemispheres, Supplemental Figure 5; response to comment #1
4. Distribution of beta values, Supplemental Figure 6; response to comment #4
5. PS values for representative subjects, sorted by episodic context, Supplemental Figure 7; response to comment #8
6. PS in HC control regions, Supplemental Figure 8; response to comment #2
7. Distribution of chi-square values for trial numbers control analysis, Supplemental Figure 9; response to comment #3
8. PS/reaction time correlations, Supplemental Figure 10; response to comment #6
9. Co-registration of subfields, Supplemental Figure 11; response to comment #7
10. Addition of reaction time differences between PS conditions, Supplemental Table 1; response to comment #6
11. Table of univariate coordinates, Supplemental Table 2; response to comment #9

In response to the reviewer's suggestion that we do a more thorough job situating the novelty of the current findings in the hippocampal subfield literature, we now include the following sections throughout the discussion:

pp. 11-12:

Studies in rodents have reported differences between spatial coding in CA1 and CA3 following changes in the environment (Lee, Rao, & Knierim, 2004; Roth, Yu, Rao, & Knierim, 2012), when learning spatial tasks such as the watermaze (Gusev, Cui, Alkon, & Gubin, 2005), in distinguishing non-spatial sequences (Farovik, Dupont, & Eichenbaum, 2010), and in contextual fear conditioning paradigms (Ji & Maren, 2008).

p. 13:

In some tasks, such as those involving spatial navigation, distinguishing competing spatial representations may be essential to correct navigation and subsequent memory performance (e.g., Kyle, Stokes, et al., 2015), and, thus, CA23DG may be pushed toward orthogonalization.

p. 14:

In one recent demonstration of the flexibility of coding schemes in CA23DG, Aly and colleagues (Aly & Turk-Browne, 2016a) asked participants to deploy attention either to objects within a context (pieces of art) or the context itself (the layout of the room). Critically, stimuli were identical between the conditions but participants' attentional state varied based on the task demands. They found that CA23DG, but not CA1, showed a greater match in its profile of activity to the task-relevant attention state. This task-driven modulation of activity was related to increased subsequent memory for incidentally encoded information (Aly & Turk-Browne, 2016a). In a related study, they replicated the finding that attention state-specific patterns of activity in CA23DG were uniquely related

to subsequent memory for the studied items (Aly & Turk-Browne, 2016b). Such findings indicate that the coding scheme in CA23DG can change according to task demands, and this can have consequences for later memory performance.

p. 14:

Both temporal and spatial details are key defining features of episodic memories (Tulving, 1984), but, until now, it was not clear how these spatiotemporal contexts are represented by the HC during retrieval. CA1 and CA23DG exhibited activity patterns that were sensitive to retrieved episodic context information in the absence of memory differences for the items themselves. The complementary characteristics of these subfields explain how we can parse our experiences into cohesive episodes, while retaining the specific details that support vivid recollection (Burgess, Maguire, & O'Keefe, 2002; Eichenbaum & Cohen, 2014; O'Reilly & Norman, 2002; Ranganath, 2010).

In addition, we now cite more of the relevant work given the less restrictive reference limit. Added citations are highlighted in the resubmission.

Specific suggestions:

1. PS values should be shown for the right hippocampal subfields as well. It is misleading to run an analysis that includes hemisphere as a factor, but then only show the data for the hemisphere that happened to show a statistical effect. It makes me wonder what is going on with the other hemisphere (very noisy, same pattern but less reliable, etc.), and some answers to that question could indicate problems with the data.

We thank the reviewer for raising these concerns. As noted above, we did not intend to mislead readers—further analyses for the right hemisphere were omitted because the interaction was driven by a larger effect of episodic context in the left hemisphere than in the right. We now include results in the right hemisphere in Supplemental Figure 5.

Although there was not a reliable ROI X Context Similarity interaction in right hemisphere ($X(2) = 1.65, p = 0.44$), we repeated all analyses that are presented in the left hemisphere in the main paper for full transparency with the reviewer. Neither CA1 ($X(1) = 0.03, p = 0.85$) nor CA23DG ($X(1) = 0.70, p = .40$) systematically differed in their representation based on an item's spatial context nor was there any interaction between the subfields ($X(1) = 0.19, p = 0.66$). When considering episodic context similarity, again neither CA1 ($X(1) = 2.99, p = 0.08$) nor CA23DG ($X(1) = 0.12, p = 0.73$) systematically differed in their representation based on an item's episodic context nor was there any interaction between the subfields ($X(1) = 1.23, p = 0.27$)

2. A related issue is that a control region would have been a nice addition (a non-hippocampal location) to show specificity of these effects to hippocampal subfields.

We now include the following:

Main text, pp. 10-11:

To show the specificity of these effects to CA1 and CA23DG, we repeated our PS analyses using entorhinal cortex and subiculum instead (Supplemental Figure 8). These regions are in the hippocampal formation, but not considered to be subfields of the hippocampus proper. For Episodic Context Similarity, neither ERC (left: $\chi^2(1) = 0.09, p = 0.77$; right: $\chi^2(1) = 0.51, p = 0.47$) nor subiculum (left: $\chi^2(1) = 0.15, p = 0.70$; right: $\chi^2(1) = 3.75, p = 0.053$) differed in their representation based on an object's episodic context. For Spatial Context Similarity, neither ERC (left: $\chi^2(1) = 0.64, p = 0.42$; right: $\chi^2(1) = 1.79, p = 0.18$) nor subiculum (left: $\chi^2(1) = 0.15, p = 0.70$; right: $\chi^2(1) = 0.08, p = 0.77$) were differentially sensitive to spatial context alone.

Supplemental Figure 8:

3. Page 8 says “Control analyses that matched the number of trials across conditions revealed consistent results”. This is an alarming statement because it both raises the possibility that the main findings could be due entirely to different trial counts across conditions and gives a vague answer to that possibility. Absolutely no details are given about the nature of these control analyses or what they showed. How is the reader to evaluate this? What was done and what counts as “consistent results”?

As noted earlier, the brevity of this description was due to the brief communication format, however, our point was to reassure the reader that the main findings are **not** due to different trial counts across conditions. We have now expanded the discussion (p. 9) of this control analysis to include the following:

A control analysis was performed to ensure that our results could not merely be explained by the number of trials in each condition since the same-video/same-house necessarily had the smallest number of trial pairs contributing to the pattern similarity analysis. To do this, we took the condition with the fewest number of trial pairs on a subject-by-subject basis and randomly selected a matching number of trial pairs from all other conditions. For example, if a participant had 152 trial pairs in the same-video/same-house condition, we

randomly selected 152 trial pairs for all other conditions by generating a random sequence of numbers of length 152 and then using this random sequence to index trial pairs for selection in the mixed modelling of pattern similarity. We repeated this simulation 1000 times and looked to see whether our observed chi-square values for the critical interaction of ROI x Context Similarity x Hemisphere interaction were significant at a threshold of $p < .05$ (Supplemental Figure 9). Across all of these random samples of trials, we always obtained a significant result.

Supplemental Figure 9:

4. The reviewer brings up two concerns here: (1) data quality and (2) appropriate fit of the model to the data. We now include the requested follow-up analyses, however, we would like to clarify how the pattern similarity analyses were done and, therefore, why we are not as concerned with the fit of the model as we might be with a traditional univariate analysis (additional clarification about PS analyses is now included in the introduction on p.2). Pattern similarity analyses utilize the same trials in each condition, such that the analysis entirely depends on *which trials are correlated with one another*. For instance, consider four recollected trials: A, B, C, D. We might use $r_{A,B}$ and $r_{C,D}$ to index pattern similarity for the Same House/Same Video Condition and $r_{A,C}$ and $r_{B,D}$ to index pattern similarity for the Different House/Different Video Condition. Pattern similarity analysis parallels analyses of neural population vectors in single-unit recording studies, in which one can observe strong, meaningful brain –behavior relationships in a population of neurons that do not have obvious response selectivity properties (see Rigotti et al., Nature, 2013; for an example of hippocampal units, see Keene et al. J. Neuroscience, 2016). Thus, the underlying assumption is that, whether the overall mean activation level is negative, positive, or

negligible, the pattern of activity across the voxels can still be informative (see Haxby et al., Science, 2001 and Kamitani & Tong, Nature Neuroscience, 2005, for prominent examples).

(a) The main analysis of subfield fMRI PS really needs to be complemented by a corresponding univariate analysis in order to allow evaluation of the data quality and therefore the quality of the PS findings.

Because the same trials are used in both conditions, it would be impossible to run a corresponding univariate analysis. However, we now include a new control analysis in the Supplemental materials (Supplemental Figure 4) that the reviewer may find useful. Subfield-specific univariate activity was estimated for recollected objects within each spatial context. Neither subfield reliably differed in its activity between the two spatial contexts (CA1: $F(1, 22) = 0.92$, $p = 0.92$; CA23DG: $F(1, 22) = 3.17$, $p = 0.09$).

If the reviewer would like to request a different univariate analysis, we would be happy to include that instead.

b) What is going on under the hood? They should show estimated impulse response functions for the hippocampal subfield ROIs.

Although, as described above, this should not impact pattern similarity, to get a better sense of how well the canonical HRF provided a good fit to the data, we re-modelled our data taking a FIR approach. Overall activity for the hippocampus is shown in the figure below. The figure reveals that the response is well-approximated by the canonical HRF, with a peak centered around 6 to 8 seconds (our TR = 2.01 seconds, therefore each FIR time point is approximately two seconds) and activity returning to baseline levels by about 16 seconds. Consistent with our univariate analysis, hippocampal activity was significantly higher for “Recollect trials” (“RHit”, red) than for than familiar hits and misses combined (“FHitsANDMisses”, teal).

We also plotted estimated responses separately for CA23DG and CA1 (now included in the paper as Supplemental Fig. 3). The data for CA23DG reveal a strong response for recollected trials, whereas the overall average response for CA1 is relatively weak. To be clear, this data is averaged across trials, voxels, and subjects, and, as such, it does not reveal any information about the multi-voxel pattern in each subfield. We address this issue in more detail in our responses below.

(c) The PS was run on estimated betas, which could vary based on how well the canonical HRF matches the subfield response, which could bias the PS analysis for subfields with relatively canonical versus relatively non-canonical responses.

We agree that this could be a critical issue for univariate analyses that depend on precise estimation of overall activity within a region. Fortunately, as we mention above, this is less of an issue for pattern similarity analyses, given that the dependent measure is not absolute activity but, rather, the relative similarity of activity patterns across a population of voxels, regardless of the overall magnitude of activity. In a recent article, Pedregosa and colleagues (Pedregosa, Eickenberg, Ciuciu, Thirion, & Gramfort, 2015 *NeuroImage*) argued that mis-specification of the GLM reduces power to observe a significant effect for both MVPA decoding models and RSA. Therefore, if the reviewer is correct that the canonical HRF is not appropriate for the subfields, it seems that we would actually be *less likely* to observe a significant effect. Instead, we observed significant pattern similarity effects in *both* CA1 and CA23DG.

However, let us consider what would be expected if CA1 and CA23DG showed exactly the same voxel pattern information, but with fundamentally different HRF shapes. Misspecification of the peak of the HRF leads to systematic biases in estimated activation (see Calhoun et al., NeuroImage, 2004). In that event, we would expect the distribution of beta values in CA1 and CA23DG to be shifted relative to one another. As shown in the response to 4e below, there is no evidence of a systematic shift in any of the subjects' data (i.e., the distributions of beta values are centered around zero for both subfields for most subjects). Thus, we do not have reason to believe that the observed difference between CA1 and CA23DG was spuriously driven by differences in the fit of the HRF.

(d) Indeed, at least one of the references supporting the idea of subfield specialization indicates different CA3/CA1 timecourses of activity (Lee, Rao and Knierim 2004). Perhaps different activity timecourses are responsible for the reported analyses?

The reviewer also brings up the issue of whether the timecourse of activity differs between the subfields and whether this could lead to the observed differences between subfields. We agree that, for any univariate fMRI analyses, valid interpretations depend on an appropriate model of neural activity (because an inappropriate model would lead to misallocation of variance). Although we cannot be sure whether the timecourses of underlying neural activity are identical in the two regions, the our trials were spaced sufficiently close together (trial duration = 3 seconds; inter-trial jitter ranging from 2-8 seconds) that we would not expect large differences in the timing of the neural response across regions and the sluggish nature of the hemodynamic response renders fairly subtle differences in the neural activity timecourse inconsequential.

In the paper from Lee, Rao and Knierim 2004 (“A double dissociation between...” *Neuron*), the authors describe a shift in the center of mass of place fields recorded in CA1 and CA3 during a free foraging task where the cues either remained stable (“standard”) or changed (“mismatch”). The critical point for comparison with the present data is that Lee et al. do not report a difference in the latency of neural firing to a single event (indeed, this would be nearly impossible in a free foraging paradigm) over a period of seconds—instead, they report a difference in the selectivity of the neural population over the span of days. Thus, the timescale of the plasticity effects seen in CA1 and CA3 is not on a scale that would influence the estimated single-trial activation measures in the current study.

(e) At very least, the beta estimates of the subfields that were analyzed should be given so the reader knows at least how well the model fit the data for these regions.

As suggested by the reviewer, we now include violin plots of the estimated beta fits and we now include these data in a supplemental figure 6 with the paper. This figure shows distributions of the betas for each participant (where width of the violin indicates the number of betas for a given value), split by ROI. These figures should enable the

reviewer and the readers of the paper to see all of the relevant data that was used to compute the pattern similarity analyses.

In addition, we statistically confirmed that the central tendency of the beta distribution for each subject did not differ by ROI using paired samples t-tests. Neither the mean ($t(22) = -1.14, p = 0.26, d = -0.24$) nor median ($t(22) = -1.67, p = 0.11, d = -0.32$) beta value differed between the ROIs. Removing s018 did not change the pattern of results.

5. Ideally, univariate contrasts would be performed among the same conditions that were submitted to PS analysis. If there are univariate activity differences among the same conditions that show PS distinctions, then this would raise the possibility that PS effects are simply reflections of different univariate activities, which changes the interpretation. Mean univariate activity could be controlled in the PS analysis if needed.

As noted above (see 4a), there appears to be a misunderstanding--in a univariate analysis, one would contrast activation across separate sets of trials. In this case, our PS analyses examine differences in correlations across *pairs* of trials. The same trials are used to in each bin, and the the key difference is in which trials are correlated with one another. For example, a trial that had been studied in the brown house in video 1 would have been one of the trial pairs for the same-house/same-video bin with other trials from brown house, video 1 and also would have been one of the members of a trial pair for a same-house/different-video comparison if it were paired with a trial from any of the other brown house videos. Because the same trials are used in each condition of the PS analyses, it would be impossible to conduct an identical univariate analysis.

6. (a) The accuracy and response time data should be given for the conditions of interest for the PS analysis, not just nonspecifically for correct/incorrect trials and response type (currently Supplementary Table 1).

We now include RT differences (in milliseconds) broken down by PS condition in Supplementary Table 1 (see new version below). All trials included in the PS analyses were “remember” hits where the source (house) was also correctly identified, therefore we cannot further break down accuracy for these conditions.

Item Recognition

Correct Responses	Mean response rate +/- SD
“R” / old items	0.68 +/- 0.17
“F” / old items	0.25 +/- 0.15
“N” / new items	0.90 +/- 0.09

Incorrect Responses	Mean response rate +/- SD
“R” / new items	0.01 +/- 0.02
“F” / new items	0.09 +/- 0.02
“N” / old items	0.07 +/- 0.04

Item Recognition by PS condition	Mean RT difference (ms) +/- SD
sameVideo_sameHouse trial pairs	0 +/- 511.94
differentVideo_sameHouse trial pairs	0 +/- 512.28
differentVideo_differentHouse trial pairs	-4.59 +/- 514.85

Spatial Context Source Memory

Response Type	Mean response rate +/- SD
“old” / old item	0.71 +/- 0.11

“new” / old item

0.26 +/- 0.10

(b) It is already suggested that accuracy varied among conditions of interest for the PS analysis, and presumably response times could have varied as well. Given existing data that factors like response time variation can yield spurious classification results (e.g, Todd et al. 2013 Neuroimage), including full control PS analysis that explicitly address accuracy and response-time variation among condition should be performed.

We sought to address the reviewer’s concern that differences in reaction times could lead to a spurious PS result in two ways. Firstly, we visually inspected our data by separately plotting the difference in RTs for each trial pair split by ROI, subject, and condition (see plots below). Taken as a whole, there is no systematic relationship between RT differences between trial pairs and the activity pattern correlation. We now include this as Supplemental Figure 10.

To further rule out the possibility that RT differences could have driven the observed effects, we re-modeled our data including a random effect of the RT difference in trial pairs as suggested by Todd et al., 2013 (they used linear regression to remove the effect of RT in their MVPA data, which should be functionally equivalent to the approach we took here). After doing this, we replicated the results previously reported: There was a significant ROI x Context Similarity x Hemisphere interaction ($X^2(2) = 13.31, p = 0.001$). Follow up analyses revealed that this was driven by a reliable interaction between ROI and Context Similarity in left hemisphere ($X^2(2) = 15.65, p < .001$). When assessing the representation of Context Similarity in our left hemisphere ROIs, we found no difference between the subfields on the basis of Spatial Context Similarity ($X^2(1) =$

0.096, $p = 0.78$) whereas we did see a significant interaction indicating differences in representation for Episodic Context Similarity ($X^2(1) = 15.47, p < .001$).

We now including the following on p 10:

An additional control analysis was performed to ensure that differences in pattern similarity on the basis of context similarity did not simply reflect differences in reaction times between conditions (Todd, Nystrom, & Cohen, 2013). As suggested by Todd et al., (Todd, Nystrom, & Cohen, 2013), we re-modeled our data including a random effect of the reaction time difference in trial pairs that went into the pattern similarity analyses. Results remained unchanged after removing the effect of reaction time differences for all comparisons of interest (see Supplemental Figure 10).

7. The methods for segmenting subfields and co-registering fMRI data to these subfields leaves much to faith. The methods section should include relevant data/images to show accurate segmentation and alignment of functional data.

We now include the following Supplemental figure 11 showing a representative co-registration:

In addition, we have added the following text on p. 19 to the fMRI acquisition and pre-processing section better explain our co-registration approach:

Co-registration between the native-space ROIs defined in T2 space and the functional images was done with SPM's Coregister: Estimate and Reslice procedure. This procedure uses the normalized mutual information cost-function between a reference (mean

functional) image and source (T2) image to compute and apply a voxel-by-voxel affine transformation matrix. This transformation matrix was then applied to the subfield ROIs that had been defined in T2 space (see ROI segmentation) to bring them into register with the functional images (see Supplemental Figure 11).

8. a) *What is the nature of the “patterns” captured by the PS analysis?*

We now include the following in the Introduction on p.2:

One way to do this is with pattern similarity (PS) analyses. We can think of PS analyses as capturing the activity across voxels that act as spatiotemporal filters sampling from blood vessels, which, in turn, respond to fluctuations in activity from large populations of neurons (Kriegeskorte et al., 2008 Frontiers in Systems Neuroscience). Voxel pattern similarity analysis is analogous to population vector analysis approaches used in single unit recordings (e.g., McKenzie, Frank, Kinsky, Porter, Rivière & Eichenbaum, 2014 Neuron). If we assume that activity in each voxel reflects the outcome of vascular sampling of neural responses from large distributed populations, then the voxel pattern is a filtered, macro-scale analog of the neural population vector. Just as different population vectors reflect changes in the relative firing rates across different neurons across two conditions, different voxel patterns may reflect changes in the underlying population-level response across two conditions.

b) *Overall, mean PS is strikingly low ($\sim r=0.003$ on average from Figure 2). What does this mean?*

The range of observed pattern similarity values is also close to that observed from other high-resolution data sets in the Ranganath (mean PS 0.005, personal communication, Liang-Tien Hsieh) and Ekstrom (0.005-0.015 in Copara et al., 2014) labs. The relatively low correlations between voxel patterns reflect the fact that there is variance in voxel patterns that is not accounted for by spatial or episodic context. This is not surprising—even if the ideal hemodynamic pattern were identical across trials associated with the same context, we would expect noise to reduce pattern similarity estimates in real fMRI data. Moreover, at the underlying neural level, it is likely that context-related information accounts for only a small proportion of the variance observed on each trial. This is to be expected, because these items differed in many dimensions (e.g., item identity, color, potential uses, etc.) beyond spatial and episodic contexts.

c) *Are the differences in PS among conditions due to just one or two voxels being more alike/different in one condition versus the other?*

In order to control for the possibility that the differences in PS among conditions is simply due to differences in a small number of voxels, we repeated our analyses dropping voxels from each ROI in three different ways (a description of this is now included in the paper on pp. 9-10). The results remained unchanged with each method for removing influential voxels, but we detail our procedures and results here for the benefit of the reviewer.

First, we took the absolute beta value for each voxel at every trial, computed the mean value for that voxel across trials, and then selected the five voxels from each ROI with the largest mean beta value. A rationale for this procedure was that voxels with the most extreme mean beta values might be likely to drive our pattern similarity correlations. After identifying these voxels, we used the identical analysis pipeline to compute pattern similarity across our ROIs and conditions of interest.

The pattern of results remained unchanged even after dropping these voxels from each ROI. We again observed a significant ROI X Context Similarity X Hemisphere interaction ($X^2(2) = 11.55, p = 0.003$) that was driven by a reliable interaction in left ($X^2(2) = 14.63, p < .001$) but not in right hemisphere ($X^2(2) = 1.11, p = .57$). To further break down this interaction, we conducted follow-up analyses and left hemisphere. Again, neither CA1 ($X^2(1) = 0.03, p = 0.87$) nor CA23DG ($X^2(1) = 0.07, p = 0.79$) systematically differed in the representation based on an item's spatial context. When considering episodic context similarity, CA1 activity patterns were more similar for pairs of items from the same as compared to different episodes ($X^2(1) = 6.49, p = 0.01$). CA23DG showed the reverse pattern such that pairs of items from the same episode were less similar than those from different episodes ($X^2(1) = 8.48, p = .004$). We observed a significant interaction ($X^2(1) = 14.46, p < .001$), indicating that even after dropping the top five voxels with the highest beta values we observed a qualitative difference in the profiles of CA1 and CA23DG for episodic context similarity.

Our second approach involved taking the standard deviation of the beta values for each voxel across all trials and selecting the five voxels from each ROI with the largest standard deviation in their value across trials. This allowed us to identify the most variable voxels, which, in turn, could drive between-conditions differences in the episodic context similarity analysis.

The pattern of results remained unchanged even after dropping these voxels from each ROI. We again observed a significant ROI X Context Similarity X Hemisphere interaction ($X^2(2) = 9.56, p = 0.008$) that was driven by a reliable interaction in left ($X^2(2) = 11.44, p = .003$) but not in right hemisphere ($X^2(2) = 1.15, p = .56$). To further break down this interaction, we conducted follow-up analyses in left hemisphere. Again, neither CA1 ($X^2(1) = 0.02, p = 0.96$) nor CA23DG ($X^2(1) = 0.30, p = 0.59$) systematically differed in the representation based on an item's spatial context. When considering episodic context similarity, CA1 activity patterns were more similar for pairs of items from the same as compared to different episodes ($X^2(1) = 4.44, p = 0.04$). CA23DG showed the reverse pattern such that pairs of items from the same episode were less similar than those from different episodes ($X^2(1) = 7.75, p = .005$). We observed a significant interaction ($X^2(1) = 11.35, p < .001$), indicating that even after dropping the top five voxels with the highest variability in their beta values we observed a qualitative difference in the profiles of CA1 and CA23DG for episodic context similarity.

As a third approach, we computed the squared differences between every voxel for all trial pairs (e.g., $[\text{voxel1}_{\text{trial1}} - \text{voxel1}_{\text{trial2}}]^2$). Then, we computed a mean difference

value for each voxel within the Different Video Same House and Same Video Same House conditions. Finally, we computed squared differences for each voxel between the two conditions of interest (e.g., voxel1_DVSH_mean – voxel1_SVSH_mean) and identified the five voxels with the largest squared difference to be dropped.

One again, the pattern of results remained unchanged even after dropping these voxels from each ROI. We again observed a significant ROI X Context Similarity X Hemisphere interaction ($X^2(2) = 10.03$, $p = 0.006$) that was driven by a reliable interaction in left ($X^2(2) = 11.18$, $p = .004$) but not in right hemisphere ($X^2(2) = 1.32$, $p = .52$). To further break down this interaction, we conducted follow-up analyses and left hemisphere. Again, neither CA1 ($X^2(1) = 0.21$, $p = 0.65$) nor CA23DG ($X^2(1) = 0.27$, $p = 0.61$) systematically differed in the representation based on an item's spatial context. When considering episodic context similarity, CA1 activity patterns were numerically more similar for pairs of items from the same as compared to different episodes although this difference was not reliable ($X^2(1) = 0.34$, $p = 0.55$). CA23DG showed the reverse pattern such that pairs of items from the same episode were less similar than those from different episodes ($X^2(1) = 19.23$, $p < .001$). We observed a significant interaction ($X^2(1) = 10.81$, $p = .001$), indicating that even after dropping the top five voxels with the greatest between condition variability we observed a qualitative difference in the profiles of CA1 and CA23DG for episodic context similarity.

This analysis shows that voxels with the most extreme responses did not drive the differences between conditions. This is not surprising, given that there were 20 different episodic contexts (10 videos per house) in the current experiment. Thus, if one or two voxels showed large differences across a few of the episodic contexts, this would not make a large contribution to the overall difference between Same Context and Different Context trial pairs.

- d) *Given the geometry of the hippocampal subfield masks, what do different patterns reflect? There is not much space in the coronal plane, so do different patterns indicate differences in activity along the long axis?*

The reviewer is correct that if our ROIs included the entire extent of the hippocampus, we might have expected differences along the long axis especially given the known differences in the distribution of CA1 and CA23DG (for example, much of anterior hippocampus is thought to be exclusively CA1). Accordingly, in-line with the best practices in the high-res field, all of the analyses presented in the paper were restricted within the body of hippocampus.

- e) *Some unpacking of the nature of PS values, including potential visualization of similar/different patterns of representative PS levels, would be very useful in understanding the nature of the reported effects.*

In order to better understand the nature of the PS values, we plotted mean pattern similarity values for each subject broken down by ROI and condition (Supplemental

Figure 7). This enables us to understand the relative levels of similar and different PS for each condition and see how this varies across subjects. If the reviewer has another suggestion for how they would like us to visualize representative PS levels, we would be happy to include an alternate figure.

9. The rationale for the current univariate analysis is unclear. Why run a univariate analysis of recollection-related activity when the main PS comparisons involve differences by context condition? The point seems to be to show that the task “engages” the hippocampus, but why showing this for memory conditions unrelated to the conditions of interest is important is unclear

(one could presumably look at things like all stimuli versus baseline to show “engagement”). If the authors keep the current univariate analysis, activity should be summarized in table form to allow thorough evaluation and to facilitate meta-analyses.

We thank the reviewer for highlighting this point of confusion. As they point out, our univariate analysis focused on recollection-related activity. A large number of studies have shown that the hippocampus shows increased activity during successful recollection, as compared with trials where recollection fails. Accordingly, the goal was to validate the quality of the data by showing that we could replicate this finding. Finding a difference against baseline would be less informative, because it is well-known that hippocampal activity can be sensitive to memory even when overall activity is below baseline levels (Stark and Squire, 2001).

As the reviewer suggests, we now include a full summary of the activity observed in this univariate analysis in supplemental table 2:

MNI x	MNI y	MNI z	Region	Hemi	cluster size (k)	t
24	-39	-15	Fusiform	R	108	7.92
32	-40	-9	Lingual gyrus	R	-	4.69
40	-73	34	Lateral occipital cortex	R	1517	7.71
46	-72	28	Lateral occipital cortex	R	-	7.4
44	-63	30	Lateral occipital cortex	R	-	6.6
-10	-58	22	Retrosplenial cortex	L	2134	7.62
12	-57	19	Retrosplenial cortex	R	-	6.93
-6	-45	39	Retrosplenial cortex	L	-	6.84
-42	-67	28	Lateral occipital cortex	L	2668	7.37
-39	-78	36	Lateral occipital cortex	L	-	6.87
-44	-55	19	Angular gyrus	L	-	6.83
-63	-9	-18	Middle temporal gyrus	L	383	7.3
-50	-15	-20	Middle temporal gyrus	L	-	6.05
-56	-16	-12	Middle temporal gyrus	L	-	5.28
6	-48	3	Posterior cingulate	R	125	6.6
4	-54	9	Precuneus	R	-	5.54
-24	-21	-20	Hippocampus	L	88	6.37
3	12	-8	Subcallosal cortex	R	254	6.28
-4	18	-5	Subcallosal cortex	L	-	5.89
0	12	1	Subcallosal cortex	-	-	5.52
-2	-49	60	Precuneus	L	210	6.07
9	-40	39	Posterior cingulate gyrus	R	-	5.08
4	-37	45	Posterior cingulate gyrus	R	-	4.48
-32	-36	-15	Fusiform	L	92	5.68
-26	-42	-9	Lingual gyrus	L	-	4.27
-18	-37	-12	Hippocampus	L	-	3.85
6	27	-11	Subcallosal cortex	R	133	5.68
3	35	-15	Frontal medial cortex	R	-	5.12
62	-57	-9	Middle temporal gyrus	R	111	5.5
64	0	-21	Middle temporal gyrus	R	89	4.88

58	-9	-24	Middle temporal gyrus	R	-	4.84
68	-7	-21	Middle temporal gyrus	R -		4.78

We might be misunderstanding the reviewer’s point about the “context condition,” because the same trials were included in all conditions of interest in the PS analysis. That is, the “same context” trial pairs were comprised of all of the trials associated with recollection responses and successful spatial context memory decisions, and “different context” trial pairs included the same trials. Thus, it is impossible to run a univariate analysis that matches the conditions of interest in the PS analysis. We apologize if we have misunderstood the reviewer’s request and we can perform any specific contrast on request.

10. One would have guessed that right hippocampus would be important for this task, but all findings were specific to left. Univariate responses for recollection also seem to be on left (Figure 1, although details on orientation of display seem missing so hard to tell if it is left or right). Was the laterality unexpected? Should additional guards against multiple comparisons be used given that left hippocampus does not seem to be an a priori hypothesis.

As we addressed in our response to item #1 above, we had no strong expectations regarding the laterality of effects. Although it is not clear why the effect of episodic context was more reliable in the left hemisphere than the right, this is not an unusual phenomenon in functional neuroimaging. The overwhelming majority of univariate standard-resolution fMRI studies that report hippocampal activation during successful memory encoding or retrieval show unilateral effects (see Diana et al., 2007 for a review; see also Eldridge, Engel, Zeineh, Bookheimer, and Knowlton, 2005 Figs 2 & 4) and this is also true of high-resolution studies (e.g., Berron et al., 2016 Figure 3A Brown, Hasselmo, and Stern 2014 Figure 3; Carr, Viskontas, Engel, and Knowlton 2009 in text description, p. 2658; Copara et al., 2014 Figure 3b; DeShetler & Rissman, 2017 Figure 3; Schlichting, Zeithamova, and Preston, 2014 Figure 3; Suthana, Ekstrom, Moshirvaziri, Knowlton, and Bookheimer, 2009 Figures 3 and 4). Previous voxel pattern analysis studies also often report unilateral effects in the hippocampus (see, for example, Kim, Norman, & Turk-Browne, 2017; Hsieh et al., 2014). The issue is not just in relation to laterality—many univariate and multivariate studies report voxel-based analyses that reveal significant, replicable effects that are sometimes limited to one part of the hippocampus (e.g., the left hemisphere, a cluster, a subfield).

The observed interaction might point to an interesting difference between the hemispheres. Although the same structures exist in both hemispheres, we know that various processes can be lateralized (e.g., language, handedness) in the human brain. In general, laterality effects are poorly understood—for example, Aron, Robbins, and Poldrack, 2014 highlight a role for the right, but not the left, inferior frontal gyrus in response selection, we know of no explanation for the laterality effect. Because our study was not intended to test hypotheses about hippocampal laterality, we did not speculate about this issue in the paper, but our revised manuscript presents the results clearly so that other interested researchers might pursue this issue further.

Regarding the statistical issue raised by the reviewer, our analysis approach was conservative, in that we included hemisphere as a factor. We followed up on the left hemisphere results, because of the observed ROI x Context Similarity x Hemisphere interaction indicated that the effects

were different in the two hemispheres. The subsequent analyses were designed to break down the cause of the observed effects in relation to ROI and Context Similarity, and, as such, these analyses were not exploratory.

11. The rationale for not performing slice-timing correction given MB acquisition is unclear. It is possible.

To the best of our knowledge, the ability to perform slice-timing correction for MB (multiband EPI) sequences was not implemented until SPM12, and the analyses for this paper were done in SPM8. In addition, there are two complex issues involved in whether to perform slice-timing corrections. First, slice acquisition is confounded with motion in time. Second, slice-timing corrections can introduce interpolation errors. One could argue that, at a 2s TR, the interpolation errors would be small, but conversely, in our experience, slice timing correction has modest effects, and we have never found it to change the overall pattern of results. There is no reason to believe that slice-timing correction would change the results of the voxel pattern analysis reported here.

12. A rationale for why mixed modeling was used rather than traditional repeated-measures ANOVA should be included. It seems like a simple 2x3x2 design, so the rationale for modeling is not clear.

Mixed modelling is routinely recommended by statisticians as a best practice approach to data analysis (Baayen, Davidson, and Bates 2008 JML; Clark 1973 JVLVB; Dixon, 2008 JML; Jaeger, 2008 JML; Singmann, Kellen, Spieler, and Schumacher, book in press), including fMRI research (e.g., Mumford & Poldrack, 2007 SCAN). The use of mixed models is not unusual in functional neuroimaging, and in fact, it is implemented as the standard analysis approach in the FSL software package.

It is important to note that ANOVA is just a special case of the General Linear Model, but the traditional “summary statistic” ANOVA approach does not account for within-subject error variance. In contrast, the mixed model approach differentiates between within-subject and across-subject error variance. Mumford and Poldrack (2007) point out that “[a] random effect does not change the mean structure of the model, but it changes the variance structure so the distribution associated with the model matches the distribution of the data.” In short, mixed modeling is a better statistical practice than ANOVA.

REVIEWERS' COMMENTS:

Reviewer #2 (Remarks to the Author):

The authors did an excellent job expanding the description of this study and the additional control analyses on trial counts, voxel subsets, and response times turn this into a convincing report. I have no other comments.